# From lab to real life: Is there a link between lab-based and ecological assessment of Procedural Perceptual-Motor Learning tasks?

**Elodie Martin** [1,2,3]*, **Sarah Seiwert**[1], **Lilian Fautrelle** [1], **Joseph Tisseyre** [1], **David Gasq**[1,2], **Martin Lemay**[4,5], **David Amarantini** [1], **Jessica Tallet**[1]

**1** ToNIC, Université de Toulouse, INSERM, UT3 Paul Sabatier University, Toulouse, France, **2** Department of Functional Physiological Explorations, Motion Analyses Center, University Hospital of Toulouse, Hopital de Purpan, Toulouse, France, **3** Institut de Formation en Psychomotricité, Faculté de santé, Paul Sabatier University (UT3), Toulouse, France, **4** CHU Sainte-Justine, Research Center, Université du Québec à Montréal, Montréal, Quebec, Canada, **5** Department of Physical Activity Sciences, Université du Québec à Montréal, Montréal, Quebec, Canada

* elodie.martin@inserm.fr

## Abstract

Procedural Perceptual-Motor Learning (PPML) refers to the process leading to the acquisition of new motor skills through repeated practice. It is crucial to (re-)acquire skills needed in daily life and rehabilitation. It can be divided in two processes: motor sequence learning (SL) and sensorimotor adaptation (SA). SL refers to the acquisition of a sequence of actions that follows a precise order, while SA involves continuously adjusting motor outputs to compensate for environmental or internal disturbances. These two processes are typically measured using different lab-based tasks and are presumed to play a role in ecological/ naturalistic tasks. However, to our knowledge, no study examined the relationship between performance on lab-based tasks and ecological/ naturalistic tasks. To address this gap, we designed two lab-based tasks and six ecological tasks assessing SL and SA in an original research including 42 participants (young adults). After ensuring with non-parametric repeated measures ANOVA that all the tasks presented features of learning (all $15.1 < \chi^2 < 142$; $p < 0.5$), Spearman's rank correlation tests were performed between each lab-based task measuring SL and SA and the six ecological tasks. Our findings reveal low to moderate correlations between lab-based and ecological tasks measuring SL and SA ($0.265 < rho < 0.395$; $p < 0.05$). This suggests that the lab-based tasks partially reflect PPML as it occurs in everyday life. We believe that the partial ecological validity of these lab-based tasks is essential for their use, especially in the context of clinical evaluation prior to rehabilitation.

## Introduction

Humans have the capacity to acquire a broad spectrum of motor skills throughout their lives, including daily activities such as tying shoelaces, riding a bike, shooting a ball, dancing, playing a musical instrument, reading, and writing. This capacity is known as Procedural Perceptual-Motor Learning (PPML) [1]. It is at the core of rehabilitative interventions,

**Data availability statement:** All relevant data are within the manuscript and its Supporting Information files.

**Funding:** This work was supported by a grant from the Fondation pour la Recherche en Psychomotricité et Maladies de Civilisation, hosted by the Fondation de France (https://frpmc.fr/) and a financial support from the Agence Nationale de la Recherche (https://anr.fr;Projet-ANR-21-CE28-0031). The funders had no role in study design, data collection and analysis, decision to publish, or preparation of the manuscript (EM).

**Competing interests:** The authors have declared that no competing interests exist.

**Abbreviations:** ERR, Error; PPML, Procedural Perceptual Motor Learning; RT, Response Time; SA, Sensorimotor Adaptation; SL, Motor Sequence Learning; SRTT, Serial Reaction Time Task; TJT, Target Jumping Task.

enabling individuals to learn or relearn procedures that enhance their adaptation to the environment and promote social participation. PPML is widely studied in the scientific literature through various tasks, from highly controlled lab-based tasks to ecological (also referred to as naturalistic) tasks that closely resemble real-world activities. In the present study, we explore the possible links between these two types of tasks, as it is crucial to determine whether measurements align in both laboratory and ecological contexts, especially for standardized evaluation preceding a rehabilitation program.

PPML refers to the internal processes that enable individuals to modify rapidly and enduringly their behavior in a new task [2]. Over time, research has distinguished two types of PPML: motor sequence learning (SL) and sensorimotor adaptation (SA) [3–8]. On one hand, SL involves combining isolated movements into a single, coherent, or smooth action, known as "chunking" [9–12]. On the other hand, SA enables the maintenance or adaptation of the accuracy of movements in the presence of environmental or internal disturbances by continuously adjusting motor outputs [13,14].

From a functional and neurobiological perspective, SL and SA rely on two distinct cortico-subcortical networks: the cortico-striatal and cortico-cerebellar loops. Fast learning occurs at the beginning of practice and is characterized by a rapid improvement of time, speed, score of accuracy or number of successful attempts. Both cerebral loops are involved during this phase. In contrast, slow learning leads to more progressive improvements [15,16]. At this stage, a specialization occurs where SL primarily engages the cortico-striatal loop while SA specifically depends on the cortico-cerebellar loop [5].

There is a great diversity of tasks measuring PPML in the literature, including highly controlled lab-based tasks reserved for laboratory research and ecological that closely resemble real-world conditions. Focusing on ecological tasks allows us to observe the acquisition of skills as they occur in real-life contexts, yielding results that can be readily generalized to daily life [17]. This approach investigates a wide range of activities, from learning daily gestures such as filling a container [17] and writing [18], to performing a word in sign language [19]. It also includes sports movements in taekwondo, dance, basketball, and tennis [20–25], musical gestures [26] and highly specialized professional gestures, such as tying a surgical knot [27–29]. However, the heterogeneity of these studies makes their results difficult to compare. As illustrated by Ranganathan et al. [30] in golf putting, paradigms can be very variable even when researchers choose the same tasks to learn. Moreover, studies measuring both SL and SA often involve tasks with significantly different perceptual and motor constraints [31–35] and use different measures.

In SL paradigms, one of the most frequently employed paradigms in scientific research is the Serial Reaction Time Task (SRTT) [36,37] in which participants learn a specific repetitive spatial sequence by pressing the corresponding button in reaction to a stimulus appearing at one of four locations on a computer screen [38]. One task evaluating online SA is the double-step pointing in which participants make a pointing movement toward a specific visual target that makes a sudden change of location after the initiation of the movement (double-step) [39,40].

Understanding the relationship between these lab-based and ecological tasks is crucial, as lab tasks provide precise control over variables, while ecological tasks more closely replicate real-world challenges. Moreover, acquiring durable new motor skills in the real world requires considerable practice, often over a long period (e.g., learning to tie shoelaces, to write or to ride a bike) while traditional lab-based tasks (such as adjusting pointing movements during prismatic exposure or learning a keypress sequence) can usually be learned in one session lasting a few minutes or hours [36,41]. Although lab-based tasks have been widely used to study motor learning, their relevance to real-world skill acquisition remains debated (see [7,42]). This gap in understanding

is critical, as it challenges the ecological validity of decades of motor learning research and its potential applications. While, ecological tasks provide a more realistic observation of how skills are acquired in a natural context, their measurements tend to be less precise than what is achieved in controlled lab-based tasks. Given the potential implications for rehabilitation, teaching, and coaching, it is important to assess whether these two types of tasks are comparable and thereby understand to what extent results from lab-based tasks are applicable to real life.

On this basis, the primary objective of the present study is to examine the relationship between lab-based and ecological measures of SL and SA. To this aim, we investigate the correlation between (1) motor sequence learning of a lab-based task (SRTT) and three ecological tasks and (2) sensorimotor adaptation of a lab-based (TJT) and three ecological tasks. We hypothesize a positive correlation within scores of lab-based and ecological tasks measuring SL and SA respectively (intra-category), but anticipate weaker or non-existent correlation between scores of tasks measuring SL and SA (inter-category). As a prerequisite, we first confirm that all the tasks showed a significant improvement of speed and/or accuracy with practice, thereby ensuring that fast learning occurred. By addressing these questions, our study aims to fill a critical gap in the motor learning literature, providing new insights into the generalizability of laboratory findings and their relevance to real-world skill acquisition.

## Method

The study took place from February 7, 2022, to April 25, 2022.

### Participants

Forty-two adults, without self-reported neurological, mental or physical disorders and with normal or corrected-to-normal vision participated voluntarily in this study (N = 42, 31 females, mean age: 25.63 ± 4.99 years old, 4 left-handed).

The experimental design adhered to the principles of the Declaration of Helsinki and received approval from the university's ethic committee (CER 2020-320, Comité d'Ethique et de Recherche de l'Université de Toulouse). All participants provided written informed consent to participate after being informed on the experimental procedures.

### Tasks

All participants completed two lab-based tasks and six ecological tasks. Participants completed five blocks per task, with instructions to prioritize speed and accuracy repeated at the start of each block. Blocks 1, 2, 3, and 5 included the learning element (sequence for SL or sensory perturbation for SA), while Block 4 excluded it (random sequence for SL or no sensory perturbation for SA). Feedback on average movement time and percentage accuracy was provided after each block.

**Lab-based tasks.** To evaluate SL and SA in lab-based conditions, we developed adapted versions of the Serial Reaction Time Task (SRTT) and a Target Jumping Task (TJT), respectively. The TJT was based on the double-step pointing paradigm [39,40]. These tasks were implemented using Open Sesame® software (version 3.3.8, https://osdoc.cogsci.nl/ [43]) on a Dell computer (17-inch screen, standard resolution of 1920x1080 pixels). Participants sat comfortably approximately 40 cm away from the screen and used a wired with their dominant hand to respond.

In both tasks, participants were presented with a stimulus represented by a bird, which could appear in one of four positions (see Fig 1A). Following three incorrect responses or a delay of 5000 ms without response, the bird disappeared, and the next stimulus was presented 200 ms later in a different location.

**A - Position of targets for SRTT and TJT**

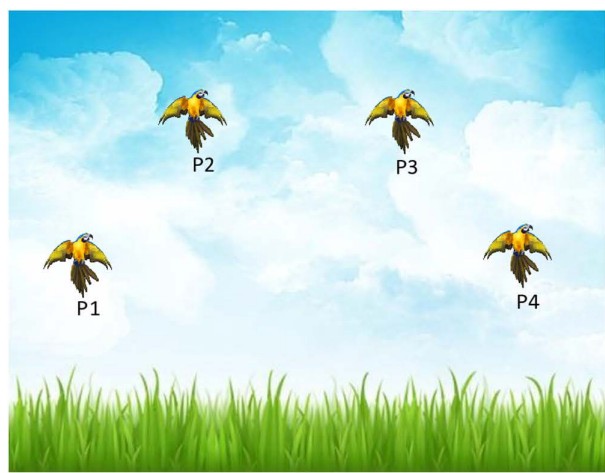

**B - Serial Reaction Time Task**

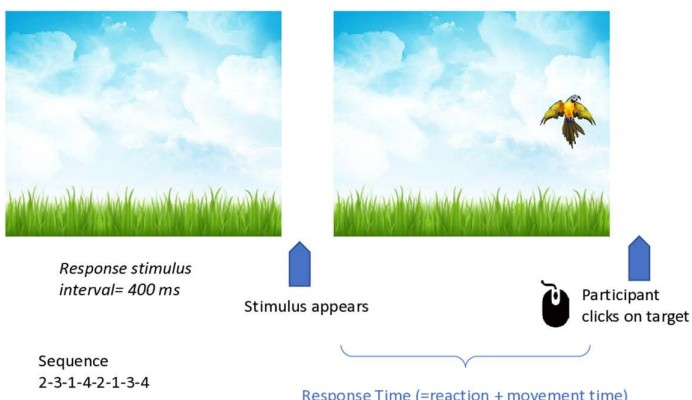

**C- Target Jump Task**

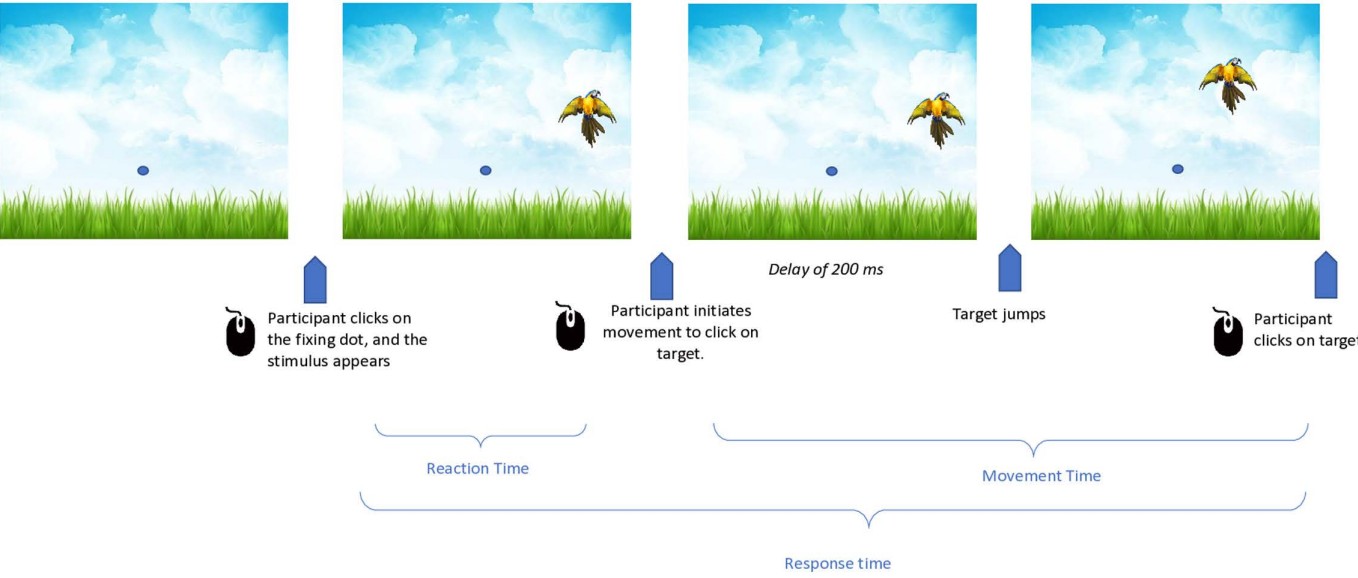

**D – Organization of practice**

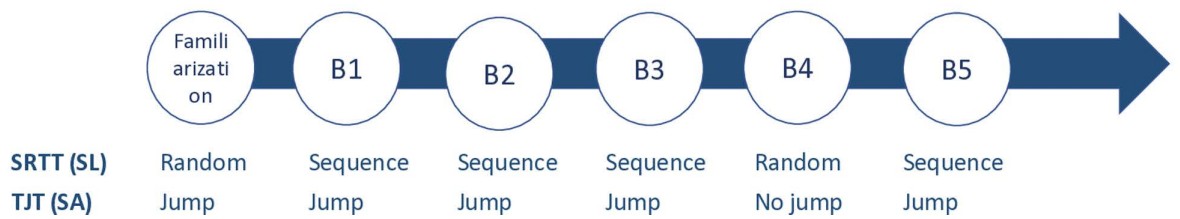

**Fig 1. Illustration of the two lab-based tasks: Serial Reaction Time Task (SRTT) and Target Jumping Task (TJT). A-** For the 2 lab-based tasks, the bird can appear in position 1, 2, 3 or 4. **B-** In the SRTT, the stimulus remained on the screen until a click response was made on the bird. When the participant click on the bird, the bird disappeared and the following bird appeared 200ms after at a different location. **C-** In the TJT, the participant was asked to click on a fixation

point in the center of the screen, and the bird appeared. 200ms after mouse left the fixation point, the bird "jumped", forcing the participant to adapt his movement. **D**- The organization of practice is similar for the TJT and SRTT, with bird appearing randomly in the SRTT and remaining stationary in the TJT during the familiarization block and block 4. Bird appears sequentially in the SRTT and with a jump in the TJT during blocks 1, 2, 3, and 5. The tasks are visible at this link: https://vimeo.com/652032612.

In the SRTT, used to measure SL, we adapted the task initially introduced by Nissen and Bullemer (1987). Instead of responding by pressing a corresponding key, participants were required to use the mouse to click directly on the stimulus. The regular patterns within the sequence are acquired in a manner similar to how participants interact with a keyboard or mouse [44]. The birds followed a predetermined sequence: 2-3-1-4-2-1-3-4, with positions 1 to 4 presented from the left to the right positions (Fig 1B). This sequence was repeated six times for block 1,2,3 and 5, resulting in a total of 48 mouse clicks. In the fourth block only, the birds appeared randomly (Fig 1D). Introducing a randomized block allows to estimate the effects of SL.

In the TJT, used to measure SA, participants were asked to click on a fixation point located in the center of the screen. Subsequently, after the participant's mouse cursor left the fixation point, a bird appeared. After a 200 ms delay, the bird "jumped" either by 25° to the right (for stimuli on the right of the screen) or 25° to the left (for stimuli on the left of the screen), prompting participants to adapt their movement during blocks 1,2, 3 and 5 (Fig 1C). Each block was composed of 24 stimuli, totaling 48 mouse clicks. Conversely, in the fourth block, the bird remained stationary (Fig 1D). Including a block without the jump allowed the assessment of the effects of sensorimotor adaptation.

**Ecological tasks.**  We developed six ecological tasks involving object manipulation to mirror real-life conditions and allowing the assessment of both SL and SA. Three tasks were primarily designed to evaluate SL and three focused on SA, as classified by a panel of eight persons including researchers and clinicians. The six tasks described below and in Fig 2 and 3 adhered to the following three key criteria:

(1)  Feasibility: Each task was achievable within a maximum of 15 minutes;

(2)  Novelty: Tasks were entirely new to participants, with no prior experience or pre-existing knowledge;

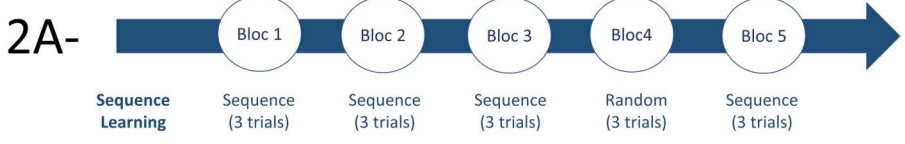

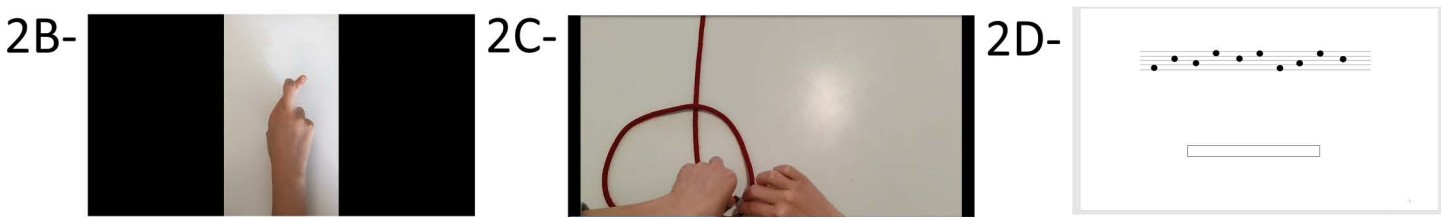

**Fig 2. Illustration of ecological tasks assessing Sequence Learning.** The tasks measuring SL are organized into 5 blocks, each containing 3 trials (2A). For blocks 1, 2, 3, and 5, the sign language word, same knot, and musical staff are presented. For block 4, participants must perform a new sign sequence, a new knot, and a musical staff. For "Learning to language word" and "Learning to tie a knot", the models are presented in first-person video format (2B and 2C). For "Learning a musical staff sequence", it is presented in PowerPoint format (2D).

(3) Minimal prior learning required: Tasks necessitate minimal or no prior learning, allowing participants to engage with them immediately (Fig 2 and 3).

Given participant's familiarity with the daily-living materials, no familiarization trials were provided for ecological tasks. All tasks were constructed in the same manner. Participants performed the tasks with their dominant hand or both hands when the tasks required it.

For SL tasks, three trials were completed in blocks 1, 2, 3, and 5, with the same sign language word, knot, and musical staff were presented. However, in the fourth block, participants were presented with a new sign sequence, a knot, and a musical staff to evaluate their capacity to perform novel tasks. In "Learning a sign language word" and "Learning to tie a knot", instructional models were presented in first-person video format (Fig 2, 2B and 2C) while "Learning a musical staff sequence" was presented in PowerPoint format (Fig 2, 1D) For SA tasks, each block contained two trials (Fig 3, 3A) with the same perturbation in blocks 1, 2, 3, and 5, (weighted pitcher (Fig 3, 3B), mirror drawing (Fig 3, 3C), and small ball (Fig 3, 3D)). However, in the fourth block, perturbations were either removed or modified (unweighted pitcher, non-mirror drawing, and large ball). The instructions always emphasized quick and accurate performance. Feedback was provided after each block regarding their speed (RT) since accuracy was directly observable.

Below is the description of the procedure for each ecological task:

*Learning a sign language word (SL_1)*: Participants sat behind a table with a computer screen displaying a video of a hand demonstrating a sign language sequence. The sequence consisted of ten elements and filmed from a first-person perspective. Two models were available: one for the left hand and one for the right hand. Participants' hands were recorded to quantify their performance.

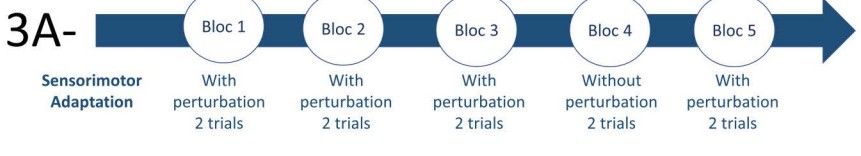

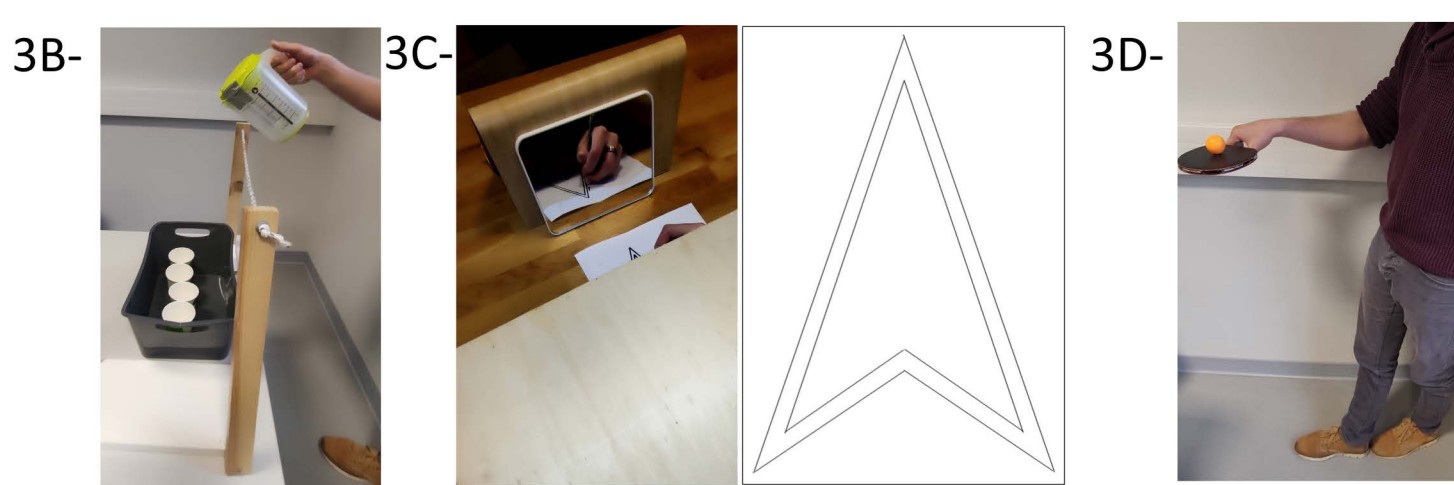

**Fig 3. Illustration of ecological tasks assessing Sensorimotor Adaptation.** The tasks measuring SA are organized into 5 blocks, each containing.2 trials (3A). "For blocks 1, 2, 3, and 5, the same perturbation is provided (weighted pitcher (3B), mirror drawing (3C), and small ball (3D)). For block 4, the perturbation is removed or modified (unweighted pitcher, non-mirror drawing, and large ball).

*Learning to tie a knot (SL_2)*: Participants sat in front of a computer screen showing a video of a hand tying a knot. The participant was required to replicate the knot while watching the video. The sequence consisted of four elements and was filmed from a first-person perspective. Participants' hands were filmed to quantify their performance.

*Learning a musical staff sequence (SL_3)*: Participants were seated in front of a computer screen that displayed a PowerPoint with a musical staff of ten points. Each point corresponded to a note linked to a computer key (four different keys: CVBN of an azerty keyboard). Participants were instructed to type their responses in a box below the musical staff. The response was displayed in white font, and participants could not read the written letter (CVBN). They were instructed not to rectify their response.

*Filling cups without spilling water (SA_1)*: Participants stood facing the table. A basin was placed on the table containing four plastic cups lined up parallel to the table. Using wooden sticks, a string was held above the basin, also parallel to the table. Participants had to pour water into the cups until they were full, while holding a weighted jug above the string without spilling any water. There was no time limit for this task.

*Mirror drawing (SA_2):* Participants sat at a table with an A5 sheet of paper placed in front of them, featuring a shape template that they had to trace with a pen. The shape template remained identical for all participants and trials (Fig 3). A wooden cover was positioned above their hands and the paper. Participants could only see their hands and the paper through a mirror placed in front of them. They had to trace the shape within two lines without exceeding them, with no time limit.

*Walking without dropping a ball (SA_3)*: Participants stood at 3-meter-long line marked on the ground. Participants started at the beginning of this line and held a racket with a ping-pong ball on it in their dominant hand, with the elbow forming an angle of approximately 90°, was demonstrated beforehand. Participants had to walk along the line heel-to-toe without dropping the ball. If the ball fell, the experimenter provided another ball, and the trial continued.

## Procedure

Participants were individually tested in a quiet room in a single session under the supervision of two experimenters. The experiment started with the administration of the Oldfield questionnaire to assess laterality [45].

Then, the two lab-based tasks (i.e., SRTT then TJT) were administrated. A familiarization block was included and participants completed a total of five blocks for each task with the same instructions to achieve the fastest and most accurate performance possible.

Finally, the six ecological tasks were administered in a randomized order. Participants received a feedback, that corresponded to the movement time of the best trial. For the ecological tasks related to SA, the accuracy of the best trial was also provided. For the ecological tasks associated with SL, scoring required subsequent notation through video analysis or in the PowerPoint presentation (see Supplementary data).

## Data

For the SL ecological tasks, errors in the quality and/or order of the elements were noted with 0.5 points awarded for correct execution, and 0.5 points for placement. For SA ecological tasks, measures included water lost in grams on each trial for the "Filling cups without spilling water" task, the number of exits and their length from the shape for the "Mirror drawing" task, the number of balls drop for the "Walking without dropping a ball task". We also measured the RT for each trial of each task. For each block, we calculated the median of response time (RT) and errors (ERR), thereby obtaining indices of speed and accuracy indices for each block and each task.

For the lab-based tasks, the software recorded the following parameters: RT for each stimulus, number of click errors before reaching the stimulus, and the position of the mouse cursor every 10 milliseconds. RT comprised the time elapsed between stimulus onset and response initiation (reaction time) and response execution time (movement time) [46]. If participants made three incorrect clicks next to the bird, the next bird was presented and the RT was noted as 5000 ms and subsequently removed from the analysis. The trajectory taken to click on each stimulus corresponds to the sum of mouse movement segments for each stimulus. The length of each segment was calculated from mouse coordinates. Aberrant coordinates (segments longer than 250 pixels) were interpolated with adjacent segments.

Due to the pronounced skewness of the errors, reaction time and trajectory distributions, we used medians that provide a more representative measure of the central tendencies of the distribution [31,47]. Thus, the median of the RT and the median of the trajectory were calculated for each block, and the number of errors per block was reported, representing speed and accuracy for each participant.

We calculated four SL scores and four SA scores based on the evolution of RT, ERR and Trajectory:

- *Global SL/SA* corresponded to the total evolution of performance during the whole task from the block 1 to the block 5

  Global SL/SA = RT (block 5) – RT (block 1); ERR (block 5) - ERR (block 1); Trajectory (block 5) – Trajectory (block 1)

- *General SL/SA* measured the evolution of performance during the first learning phase, from the block 1 to the block 3. For SL, this reflects the acquisition of visuo-motor mapping between the position of the visual cue and the required response and motor sequence [37]. For SA, it reflects an adaptive response to disturbance [48].

  General SL/SA = RT (block 3) – RT (block 1); ERR (block 3) - ERR (block 1); Trajectory (block 3) – Trajectory (block 1)

- *Specific SL/SA* measured the evolution of performance between the random/no perturbation trial blocks (block 3) and sequential/perturbation trials block (block 4). For SL, this measure typically re-increases when the sequential order is replaced by a random sequence [37,49]. For SA, the disturbance is eliminated during this de-adaptation phase, resulting in a decrease in RT/ERR/Trajectory.

  Specific SL/SA= RT (block 4) – RT (block 3); ERR (block 4) - ERR (block 3); Trajectory (block 4) – Trajectory (block 3)

- *Retention* measured the evolution of performance between blocks of sequential/perturbation trials before and after the random/no perturbation trial blocks (blocks 3 and 5). This measure assesses the possible interference induced by block 4 [38].

  Retention = RT (block 5) – RT (block 3); ERR (block 5) - ERR (block 3); Trajectory (block 5) – Trajectory (block 3)

## Statistical analyses

Data were missing for one participant in two ecological tasks (i.e., "Filling cups without spilling water" and "Walking without dropping a ball") because of task recording and for SRTT of another participant due to a computer issue.

Statistics were conducted using Jamovi® (https://www.jamovi.org/). None of the variables met the assumptions of normality (Shapiro–Wilk test, α = 0.05) and of homogeneity of variance (Levene's test, all p > 0.05). Therefore, non-parametric repeated measures ANOVA (Friedman test) were used to assess blocks effects for each task (block 1, block 2, block 3, block 4, block 5) on RT, ERR, and trajectory. Post-hoc tests (Wilcoxon rank) were then conducted to compare group pairs and identify significant differences among them. Adjusted p-values for multiple comparisons were calculated using the procedure step-by-step proposed by Hochberg (1988). To complete the information provided by the P-values, the effect sizes (ES) were calculated for each comparison [50]. To interpret ES, we rely on Cohen's approach, which suggests interpreting .10 as a small effect, .30 as a medium effect and .50 as a large effect size [51].

Spearman's rank correlation tests were conducted to evaluate the relation between SL/SA scores of SRTT and TJT, and between SL/SA scores of lab-based and ecological tasks. One-tail H1 indicated a positive correlation. Here, we used also an adjusted p-values for multiple comparisons using the procedure step-by-step proposed by Hochberg [52]. To interpret coefficient of correlation, we rely on Akoglu [53], who suggests interpreting r < .40 as a weak effect, r between .40 and .70 as a moderate effect and r > .70 as a strong effect.

## Results

### Improvements of performance in the lab-based and ecological tasks

**Lab-based tasks.** *SRTT* The ANOVA revealed a Block effect on RT ($\chi^2$ = 28,4, ddl = 4, $p$ < 0.001). Post-hoc analyses showed that the RT in block 1 was significantly higher than the TR in block 3 ($W$ = 735, $p$ < 0.001, ES = 0.707) and block 5 ($W$ = 725, $p$ < 0.001, $ES$ = 0.683). The RT in block 4 was significantly higher compared to the RT in block 3 ($W$ = 623, $p$ = 0.013, ES = 0.447) and in block 5 ($W$ = 582, $p$ = 0.050, ES = 0.352) (Fig 4). No statistical effect was observed in the ERR. The ANOVA revealed a Block effect on Trajectory ($\chi^2$ = 39,3, ddl = 4, $p$ < 0.001). Post-hoc analyses showed that the Trajectory in block 1 was significantly longer than the Trajectory in block 3 ($W$ = 724, $p$ < 0.001, ES = 0.682) and block 5 ($W$ = 788, $p$ < 0.001, ES = 0.830). The Trajectory in block 4 was significantly longer compared to the Trajectory in block 3 ($W$ = 270, $p$ = 0.037, ES = 0.373) and block 5 ($W$ = 738, $p$ < 0.001, ES = 0.714) (Fig 4).

*TJT* The ANOVA revealed a Block effect on RT ($\chi^2$ = 102, ddl = 4, $p$ < 0.001). Post-hoc analyses showed that the RT in block 1 was significantly higher than the RT in block 3 ($W$ = 778, $p$ < 0.001, ES = 0.723) and block 5 ($W$ = 797, $p$ < 0.001, ES = 0.764). The RT in block 3 was significantly higher compared to the RT in block 4 ($W$ = 903, $p$ < 0.001, ES = 1.000) while RT in block 5 was significantly higher compared to the RT in block 4 ($W$ = 903, $p$ < 0.001, ES = 1.000) (Fig 4). No statistical effects were observed in the ERR. The ANOVA revealed a Block effect on the Trajectory ($\chi^2$ = 94,6, ddl = 4, $p$ < 0.001). Post-hoc analyses showed the Trajectory in block 1 was significantly longer than the Trajectory in block 3 ($W$ = 769, $p$ < 0.001, ES = 0.703), and block 5 ($W$ = 802, $p$ < 0.001, ES = 0.776). The Trajectory in block 3 was significantly longer compared to the Trajectory in block 4 ($W$ = 892, $p$ < 0.001, ES = 0.974). The Trajectory in block 5 was significantly longer compared to the Trajectory in block 4 ($W$ = 894, $p$ < 0.001, ES = 0.980) (Fig 4).

**Ecological tasks.** *Learning a sign language word (SL_1)*: The ANOVA revealed a Block effect on RT ($\chi^2$ = 132.0, ddl = 4, $p$ < .001). The RT in block 1 was significantly higher than the RT in block 3 ($W$ = 859.0, $p$ < .001, ES = 0.903) and block 5 ($W$ = 898.0, $p$ < .001, ES = 0.989). The RT in block 4 was significantly higher than the RT in block 3 ($W$ = 901.0, $p$ < .001, ES = 0.996) and block 5 ($W$ = 903.0, $p$ < .001, ES = 1.000) (Fig 5).

The ANOVA revealed a Block effect on the ERR ($\chi^2$ = 37.5, ddl = 4, $p$ < .001). The ERR in block 1 was significantly higher than the ERR in block 3 ($W$ = 491, $p$ < .01, ES = 0.557) and

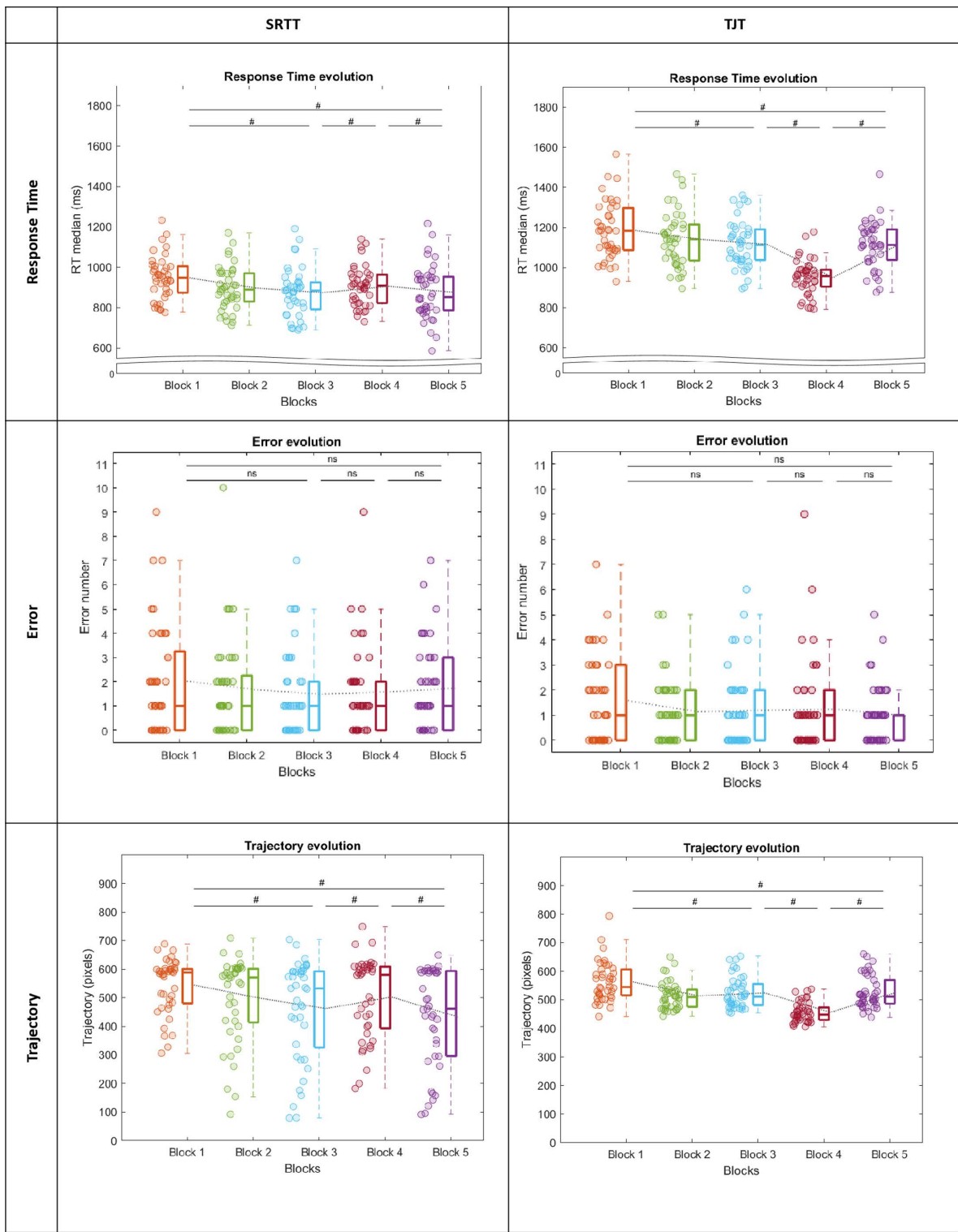

**Fig 4. Response Time, Error and Trajectory in lab-based tasks (SRTT on the left and TJT on the right).** Within each box, horizontal lines denote median values; boxes extend from the 25th to the 75th percentile of each group's distribution of values; vertical extending lines denote adjacent values (i.e., the most extreme values within 1.5 interquartile range of the 25th and 75th percentile of each group). The light gray curve represents the average for each block. The # indicate a significant difference post-hoc comparison among groups.

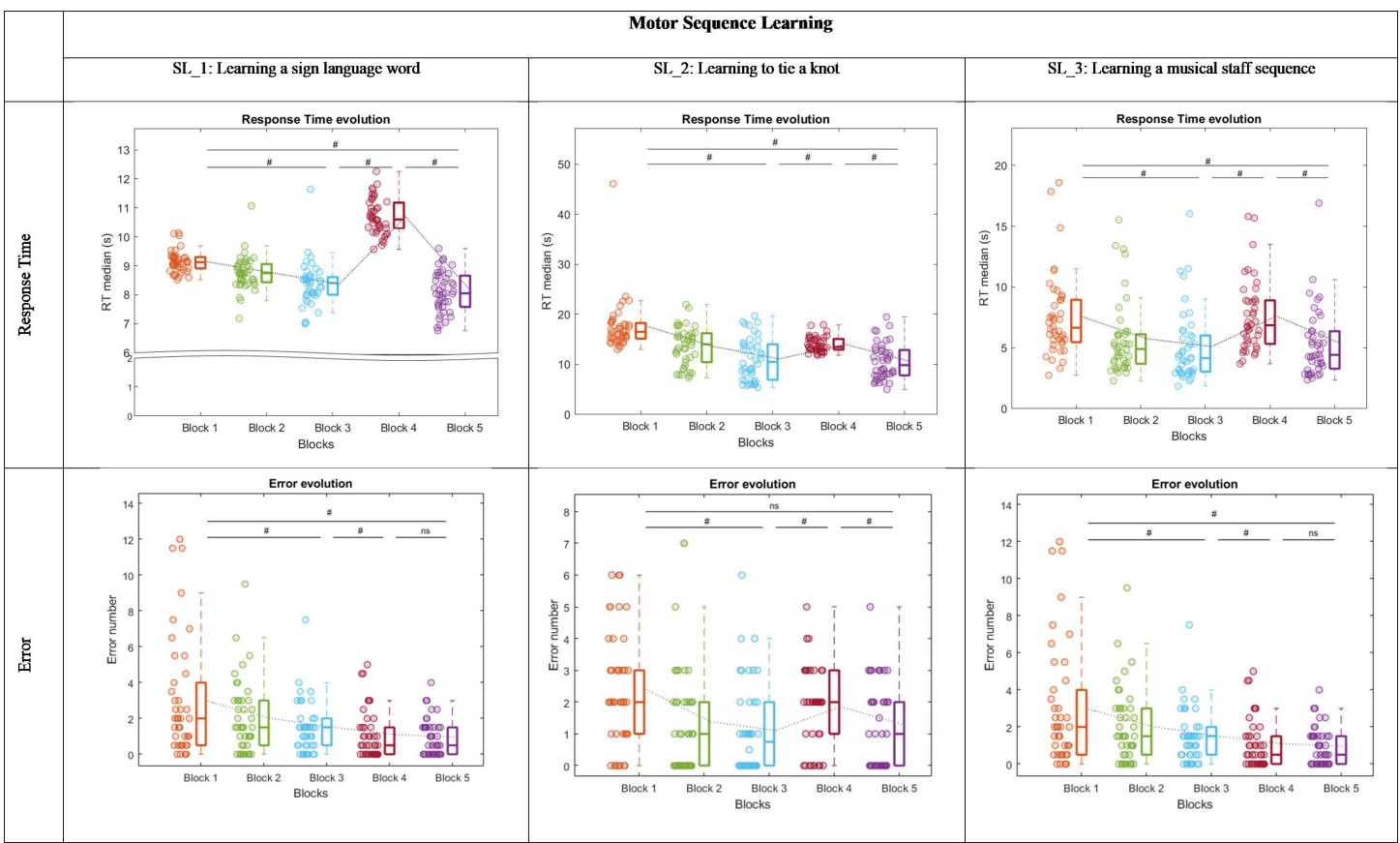

**Fig 5. Response Time and Error in Ecological Tasks assessing Sequence Learning.** Within each box, horizontal lines denote median values; boxes extend from the 25th to the 75th percentile of each group's distribution of values; vertical extending lines denote adjacent values (i.e., the most extreme values within 1.5 interquartile range of the 25th and 75th percentile of each group). The light gray curve connects the average for each block to illustrate the evolution of performance in relation to the theoretical profile.

block 5 ($W$ = 589, $p$ < .001, ES = 0.768). The ERR in block 3 was significantly higher than the ERR in block 4 ($W$ = 464, $p$ = .040, ES = 0.391) (Fig 5).

*Learning to tie a knot (SL_2):* The ANOVA revealed a Block effect on RT ($\chi^2$ = 85.9, ddl = 4, $p$ < .001). The RT in block 1 was significantly higher than the RT in block 3 ($W$ = 870, $p$ < .001, ES = 0.927) and B5 ($W$ = 892, $p$ < .001, ES = 0.976). The RT in block 4 was significantly higher than the RT in block 3 ($W$ = 750, $p$ < .001, ES = 0.660) and block 5 ($W$ = 800, $p$ < .001, ES = 0.772) (Fig 5).

The ANOVA revealed a Block effect on ERR ($\chi^2$ = 26.5, ddl = 4, $p$ < .001). The ERR in block 1 was significantly higher than the ERR in block 3 ($W$ = 536, $p$ < .001, ES = 0.700) and higher than the ERR in block 5 ($W$ = 467, $p$ < 001, ES = 0.663). The ERR in block 4 was significantly higher than the ERR in block 3 ($W$ = 411, $p$ = .001, ES = 0.657) and block 5 ($W$ = 401, $p$ = 0.029, ES = 0.772) (Fig 5).

*Learning a musical staff sequence (SL_3):* The ANOVA revealed a Block effect on RT ($\chi^2$ = 98.2, ddl = 4, $p$ < .001) The RT in block 1 was significantly higher than the RT in block 3 ($W$ = 863, $p$ < .001, ES = 0.911) and block 5 ($W$ = 848, $p$ < .001, ES = 0.878). The RT in block 4 was significantly higher than the RT in block 3 ($W$ = 878; $p$ < .001, ES = 0.945) and block 5 ($W$ = 877, $p$ < .001, ES = 0.941) (Fig 5).

The ANOVA revealed a Block effect on ERR ($\chi^2$ = 15.1, ddl = 4, $p$ =.005). The ERR in block 4 was significantly higher than the ERR in block 5 ($W$ = 420.0, $p$ = 0.004, ES = 0.589) (Fig 5).

*Filling cups without spilling water (SA_1)*: The ANOVA revealed a Block effect on RT ($\chi^2$ = 116, ddl = 4, $p$ <.001). The RT in block 1 was significantly higher than the RT in block 3 ($W$ = 861, $p$ <.001, ES = 1.000) and block 5 ($W$ = 861, $p$ =.001, ES = 1.000). The RT in block 3 was significantly higher than the RT in block 4 ($W$ = 793, $p$ <.001, ES = 0.842) (Fig 6). No statistical effects were found in the ERR.

*Mirror drawing (SA_2)*: The ANOVA revealed a Block effect on RT ($\chi^2$ = 142, ddl = 4, $p$ <.001). The RT in block 1 was significantly higher than the RT in block 3 ($W$ = 902, $p$ <.001, ES = 0.998) and block 5 ($W$ = 860, $p$ <.001, ES = 0.905). The RT in block 3 was significantly higher than the RT in block 4 ($W$ = 903, $p$ <.001, ES = 1.000). The RT in block 5 was significantly higher than the RT in block 4 ($W$ = 903, $p$ <.001, ES = 1.000). The ANOVA revealed a Block effect on ERR ($\chi^2$ = 64.9, ddl = 4, $p$ <.001). The ERR in block 3 was significantly higher than the ERR in block 4 ($W$ = 777, $p$ <.001, ES = 0.992). The ERR in block 5 was significantly higher than the ERR in block 4 ($W$ = 872, $p$ <.001, ES = 0.931) (Fig 6).

*Walking without dropping a ball (SA_3)*: The ANOVA revealed a Block effect on RT ($\chi^2$ = 112, ddl = 4, $p$ <.001). The RT in block 1 was significantly higher than the RT in block 3 ($W$ = 857, $p$ <.001, ES = 0.990) and block 5 ($W$ = 858, $p$ <.001, ES = 0.993). The RT in block 3 was significantly higher than the RT in block 4 ($W$ = 752, $p$ <.001, ES = 0.834) (Fig 6).

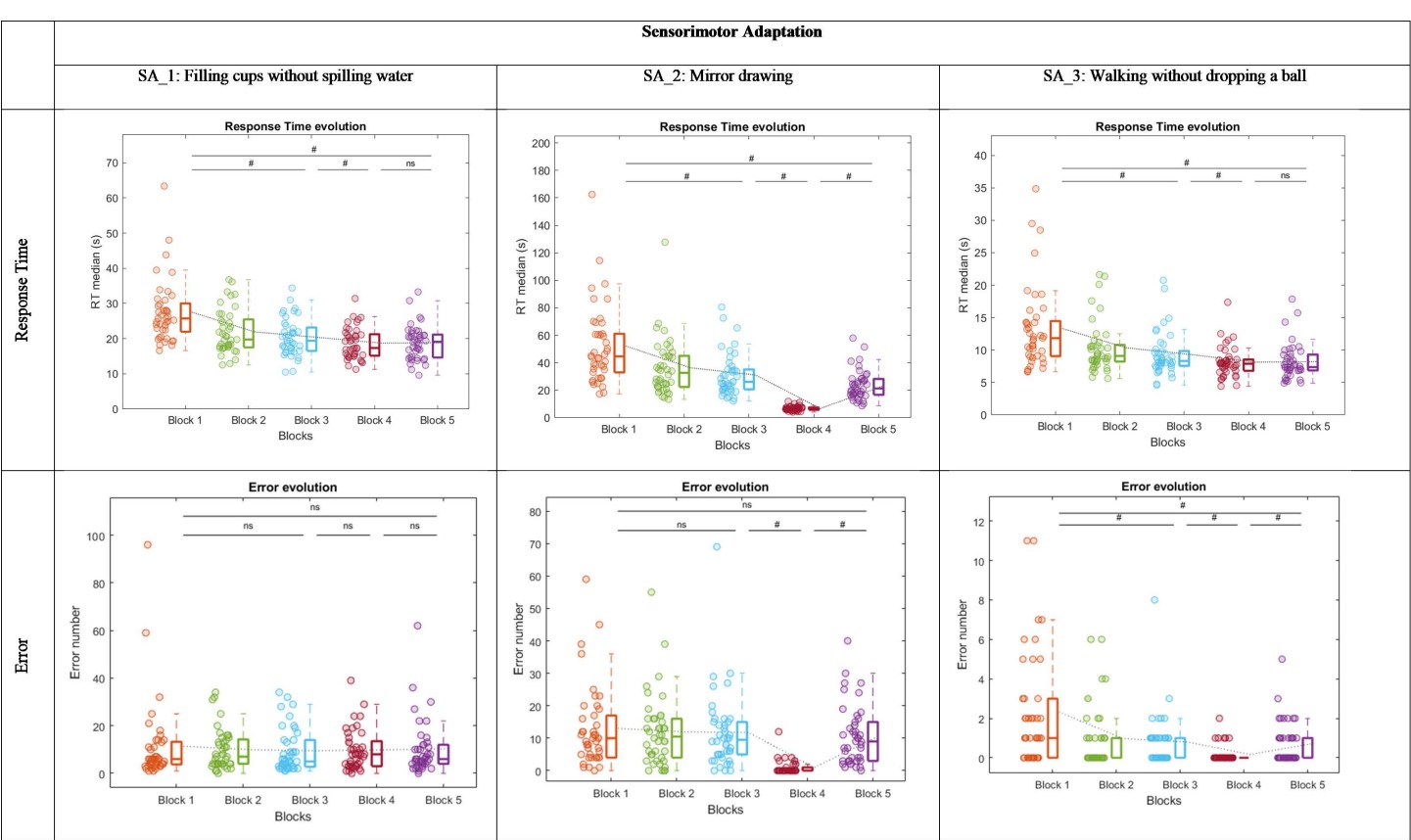

**Fig 6. Response Time and Error in Ecological Tasks assessing Sensorimotor Adaptation.** Within each box, horizontal lines denote median values; boxes extend from the 25th to the 75th percentile of each group's distribution of values; vertical extending lines denote adjacent values (i.e., the most extreme values within 1.5 interquartile range of the 25th and 75th percentile of each group). The light gray curve connects the average for each block to illustrate the evolution of performance in relation to the theoretical profile.

The ANOVA revealed a Block effect on ERR ($\chi^2 = 40,2$, ddl = 4, $p < .001$). The ERR in block 1 was significantly higher than the ERR in block 3 ($W = 482$, $p < .001$, ES = 0.718) and block 5 ($W = 228$, $p < .001$, ES = 0.970). The ERR in block 3 was significantly higher than the ERR in block 4 ($W = 188$, $p = .001$, ES = 0.786). The ERR in block 5 was significantly higher than the ERR in block 4 ($W = 165$, $p = .004$, ES = 0.737) (Fig 6).

### Correlations between results of the lab-based and ecological tasks

Since accuracy remains consistent across all tasks over time, only RT emerges as a relevant variable in both lab-based and ecological tasks. Therefore, we exclusively report analyses of RT correlations.

There were significant correlations between SL/SA scores for ecological tasks and SA/SL scores for lab-based tasks (Table 1), but no correlation was observed between the SA/SL scores of the two lab-based tasks (SRTT and TJT).

Analyses revealed a significant positive correlation between the score of global SL of the SRTT and the score of global SL of the "Learning a sign language word" ($rho = 0.395$, $p = 0.005$), and a positive correlation between score of global SA in TJT and score of global SL in "Learning a sign language word" ($rho = 0.365$, $p = 0.009$), between score of global SA in TJT and score of global SA in "Filling cups without spilling water" ($rho = 0.273$, $p = 0.042$). Higher scores of general SA in TJT were correlated with a higher score of general SA in "Filling cups without spilling water" ($rho = 0.286$, $p = 0.035$). Higher scores of specific SL in SRTT were associated with a higher score of specific SL in "Learning a sign language word" ($rho = 0.287$, $p = 0.035$). The retention score in SRTT was correlated with the retention score in "Learning to tie a knot" ($rho = 0.365$, $p = 0.009$), while the retention score in TJT was correlated with the retention score in "Mirror drawing" ($rho = 0.265$, $p = 0.045$).

## Discussion

Procedural Perceptual-Motor (PPML) can be assessed with tasks evaluating motor sequence learning and sensorimotor adaptation (SL and SA, respectively) [42]. Although authors highlight the need for standardized tasks to compare results across studies, it is essential to first establish if these tasks accurately measure motor learning as it occurs in real-life situations. That's why the present study aimed to investigate whether SL and SA measured by lab-based tasks correlate with SL and SA of the ecological tasks. We hypothesized a positive correlation

**Table 1. Correlations between SA/SL scores in RT of lab-based tasks (SRTT and TJT) and ecological tasks (SL_1: Learning a sign language word, SL_2: Learning to tie a knot, SL_3: Learning a musical staff sequence, SA_1: Filling cups without spilling water, SA_2: Mirror drawing, SA_3: Walking without dropping a ball).** $*p < 0.05$; $**p < 0.01$; N = 42, N = 41 or N = 40; Spearman rank, one-tailed. $H_a$ is a positive correlation. The grey areas correspond to the correlations attempted in our hypotheses.

| | Global SL/SA | | General SL/SA | | Specific SL/SA | | Retention | | Number of participants for each correlation for all SL/SC | |
|---|---|---|---|---|---|---|---|---|---|---|
| | SRTT | TJT | SRTT | TJT | SRTT | TJT | SRTT | TJT | SRTT | TJT |
| SL_1 | **0.395**** | **0.365**** | 0.232 | 0.151 | **0.287*** | 0.186 | 0.145 | 0.288 | N = 41 | N = 42 |
| SL_2 | 0.238 | 0.063 | −0.005 | 0.150 | 0.068 | 0.035 | **0.365**** | **0.340**** | N = 41 | N = 42 |
| SL_3 | 0.069 | −0.289 | 0.053 | −0.326 | −0.112 | −0.101 | −0.035 | −0.134 | N = 41 | N = 42 |
| SA_1 | 0.004 | **0.273*** | −0.078 | **0.286*** | −0.054 | −0.041 | 0.246 | 0.165 | N = 40 | N = 41 |
| SA_2 | 0.141 | 0.148 | 0.105 | 0.121 | −0.005 | 0.117 | −0.213 | **0.265*** | N = 41 | N = 42 |
| SA_3 | −0.057 | −0.104 | 0.072 | 0.079 | −0.035 | −0.010 | 0.135 | 0.088 | N = 40 | N = 41 |

within lab-based and ecological tasks measuring SL and SA respectively, but not between tasks measuring SL and SA.

## Ecological and lab-based tasks show signs of improvement (fast learning)

An essential prerequisite for this study was to confirm that all the tasks induced improvements of the temporal and/or spatial accuracy, in order to verify that fast learning occurred. Our findings showed consistent improvements in RT and/or accuracy (ERR/trajectory) with moderate to large effect size in both lab-based and ecological tasks [50]. For tasks measuring SL, RT and/or ERR/trajectory decreased steadily, increased during randomization, and decreased again when the sequence was reintroduced, aligning with Robertson's framework [37,54] and suggesting that participants internalized regularities of the sequence [55]. For the tasks measuring SA, RT and/or ERR/trajectory also decreased across the first blocks, approaching the baseline level of the sensory-unmodified condition, which is consistent with an improved motor performance during adaptation [4,8]. RT can reflect the participant's ability to anticipate the sensory modification (i.e., jump) and to quickly respond (to the target after its displacement) [56].

In both the SRTT or the TJT, participants demonstrated a very low incidence of errors, rarely clicking outside the target. This result is consistent with those previously found in the traditional version of the SRTT where participants make very few of them [37], contrary to SA that use the evolution of kinematic variables to measure learning [56, 57]. In our protocol, given that we wanted to align motor and sensory constraints in the two lab-based tasks, we implemented a mouse-based response. Consequently, both tasks involved continuous action, allowing us to consider the mouse trajectory as a more reliable index of learning. In both the SRTT or the TJT, participant's trajectory lengthened as they made corrections. Conversely, as their ability to anticipate the target's location improved, their trajectory became more optimal, resulting in fewer corrections and shorter trajectories. Hence, the trajectory length followed a similar evolution as the RT for both tasks, indicating that the trajectory can be a robust indicator of SL and SA.

Regarding the ecological tasks measuring SL, our results reveal participants made very few errors (with an average of 0.4/10 for "Learning a sign language word", 0.4/4 for "Learning to tie a knot", and 0.34/10 for "Learning a musical staff sequence") and the number of errors remained relatively stable over time. In all three tasks, it seems that participants prioritized accuracy over speed. Regarding ecological tasks measuring SA, the speed-accuracy trade-off seems to differ across tasks, hence reflecting that participants focused more on speed or more on accuracy as a function of the task [15,58]. For instance,. Nevertheless, the reduction of RT or ERR testify that fast learning occurred, consistent with existing research on SL and SA in more "naturalistic" tasks, which emphasizes a gradual decay in spatial or temporal variables over blocks [59–61].

## Ecological and lab-based tasks are linked

A key finding of the present study is the correlations between the SA/SL indices of the lab-based and ecological tasks measuring SL (SRTT and "Learning a sign language word", SRTT and "Learning to tie a knot") and SA (TJT and "Filling cups without spilling water", TJT and "Mirror drawing"). To our knowledge, correlations between lab-based tasks and ecological tasks have primarily been investigated in the context of neuropsychological tests to assess the validity of these measurement tools. Studies demonstrated moderate correlations between cognitive tests and the performance of everyday tasks [62–65]. We found correlations comparable to these findings, although our study was conducted in neurologically intact samples.

These correlations suggest common underlying processes in both lab-based and ecological tasks within each category: SL or SA [66–68]. Learning in the SRTT was correlated with learning in the sign language task, and learning in the knot-tying task. These results comfort that these tasks share the common process of "chunking". In contrast, learning in the TJT was correlated with learning in the Filling Cups without Spilling task, and with the Mirror Drawing task. These results comfort that these tasks require sensory error correction [69]. The involvement of these distinct mechanisms may explain the intra-category correlations that we found.

Of significant importance, despite SL and SA require common motor and perceptual processes and both require predictions of spatial events [67, 68], no correlation was found between the SA/SL indices in the SRTT and TJT. This suggests that the processes involved in these two tasks (SL and SA respectively) are relatively independent. This result extends that of Stark-Inbar et al. [32] revealing no correlation between the learning measures of the different tasks. In our protocol, the instructions and perceptual-motor characteristics are identical across our tasks. The only differences lie in the order sequential of stimuli appearance in the SRTT and in the jumps in the TJT. Therefore, we can infer that the only distinction between the two tasks was the type of PPML involved. Our results thus suggest the absence of a singular process in PPML but confirm that SL and SA are two distinct PPML processes [16], respectively measured by SRTT and TJT.

In some ecological tasks ("Learning a sign language word" and "Learning to tie a knot"), inter-category correlations were found between scores of ecological tasks involving SL and the TJT ("Learning a tie knot" and "Learning a sign language word"). It is likely that both SL and SA play a role in these ecological tasks. For instance, in the" Learning a tie knot" task, there may be an adaptative component where participants need to adjust to the rope's weight and flexibility while also ensuring correct rope placement in each trial. Regarding the link between the TJT and "Learning a sign language word", it is important to note that participants watched videos of the sequence to be learned and were asked to reproduce them as they watched. Our ecological tasks have been classified by eight researchers and clinicians as predominantly involving either SL or SA. However, this classification may not fully account for the possibility that some tasks may integrate aspects of both types of learning. Our results show that this is the case given that our hypotheses are not fully validated. No link was found between the SRTT and the "Learning a musical staff sequence" task or between the TJT and the "Walking without dropping a ball" task. Several potential explanations warrant exploration. Firstly, different strategies could have been adopted during some sequential ecological tasks. For instance, during "Learning a musical staff" sequence, it was obvious that some participants relied on counting the lines on the staff and employed declarative memorization techniques. These tasks still require significant cognitive control, making them cognitively demanding [70]. Similarly, in "Learning to tie a knot", previous research has demonstrated a strong association between mental imagery ability and knot learning [27]. Cognitive strategies may have decreased the possibility of correlation with the SRTT. Also, the absence of correlation between the "Walking without dropping a ball" task and the TJT may be attributed to the unique nature of this task. While we primarily selected tasks involving manual motor skills, the "Walking without dropping a ball" task requires broader error detection and correction, including adjustments in hand position and racket angles, as well as broader considerations such as posture, displacements, and balance maintenance. This complexity in error detection and correction may differ significantly from tasks like "Mirror drawing" (where participants had likely no prior experience) or "Filling cups without spilling water" task (where participants had similar prior experience). The variability in individuals' prior experience could also influence how errors are detected and corrected in the "Walking without dropping a ball" task thus explaining the absence of connection between the TJT and this task.

## Limitations

Despite promising results, our study has some limits. Firstly, the limited sample size (N = 42) and a larger number of females (N=31) warrants generalization of the results. It is possible that the limited sample size explains that correlations are low to moderate (0.265 < rho < 0.395). As regards to a possible gender difference, it has not been reported in the literature on PPML.

Secondly, the lab-based tasks entail precise measurements captured at the millisecond and pixel levels, while the ecological tasks have been measured using a timer for timing and grids for error assessment. To increase the accuracy of measurement, the two assessors (EM and SS) used standardized correction grids (see supplementary data) and underwent training together to ensure the most standardized correction possible. However, measurements of the ecological tasks are inherently more macroscopic than those of the lab-based tasks.

Thirdly, this work is part of a broader objective aimed at assessing the ecological validity of lab-based tasks for clinical testing purposes [71]. Thus, we standardized their administration, always in the same order. This standardization is not prone to influence our conclusions since our goal is not to compare them and there is no correlation between them.

Forty, we are presently unable to determine the persistence of learning due to the relatively short duration of the single training session (fast learning). The retention block was administered after only a short delay, during which an interference block was introduced. Investigating the correlation between the tasks after a longer delay could provide an interesting avenue for exploring their connections in a more specific manner for each type of learning.

## Conclusion

The current study is in line with Krakauer et al. [6] who propose that: "*although simple learning tasks may not enable us to fully understand how complex, real-world skills are learned, these tasks do provide foundational insights into components of learning that are likely necessary (if not sufficient) to account for how more impressive skills are learned*" (page 614). The present study bring new cues for a holistic view of how individual behavior in controlled lab environments corresponds to real-life experiences.

To assess SL and SA, lab-based tasks have their own limitations and may not always be suitable for all research situations. In situations where the goal is to understand human behavior in real-life contexts and to account for factors that may be overlooked in a laboratory environment, ecological tasks may be preferred. The choice between lab-based or ecological tasks depends on the research objectives, the nature of the questions being addressed and the available resources. In many cases, a combination of lab-based and more ecological tasks can provide a more comprehensive and nuanced understanding. However, the benefits of lab-based tasks offer distinct advantages over more ecological tasks:

- Lab-based tasks allow specific and independant measurement of SL and SA

- Lab-based tasks have minimal reliance on specialized equipment in our case, yet offer precise measurement while being not time-consuming, making them easily replicable

- Lab-based tasks allow more precise measurement of lab-based variables, both in terms of speed and accuracy.

- Lab-based tasks can be relevant tasks to explain how the motor skills are acquired in real life.

- Lab-based tasks are often designed to be repeatable, enabling researchers to consistently reproduce the experiment to validate results.

- Lab-based tasks enable minimizing interference from unwanted variables such as the participant's expertise or familiarity with related tasks.

- Lab-based tasks can be carried out more quickly, as they typically demonstrate SL and SA within a shorter timeframe compared to ecological tasks.

Even if the lab-based tasks described here could be improved, our results suggest that they are relevant for evaluating participant's ability to learn new perceptual-motor skills, both in research and in clinical practice. Our findings contribute to assess the ecological validity of lab-based tasks to transform them into clinical tests [68]. They also reinforce the foundation for the development of targeted interventions and personalized treatment strategies based on individual profiles, thereby fostering more effective and tailored approaches in both research and clinical settings.

## Supporting information

**S1 Table. Scoring accuracy in tasks measuring SL.**
(DOCX)

## Acknowledgments

The authors are grateful to the volunteers who participated in this study. The authors would like to thank Claire Cherrière, Marie Cécile Dauger, Anthony Guedes, Claire Lebely, Déborah Méligne for their participation in classifying ecological tasks according to the type of learning (SL vs SA). They also thank Louise Tison for her participation to the recruitment of participants. They thank Déborah Méligne for her role in the regulatory aspects of the project.

## Author contributions

**Conceptualization:** Elodie Martin, Lilian Fautrelle, David Amarantini, Jessica Tallet.

**Data curation:** Elodie Martin, Sarah Seiwert, Jessica Tallet.

**Formal analysis:** Elodie Martin, David Gasq, David Amarantini, Jessica Tallet.

**Funding acquisition:** Elodie Martin, Jessica Tallet.

**Investigation:** Elodie Martin, Sarah Seiwert.

**Methodology:** Elodie Martin, Lilian Fautrelle, Martin Lemay, David Amarantini, Jessica Tallet.

**Project administration:** Jessica Tallet.

**Software:** Elodie Martin, Sarah Seiwert, Joseph Tisseyre, David Amarantini.

**Supervision:** David Amarantini, Jessica Tallet.

**Visualization:** Elodie Martin, David Amarantini, Jessica Tallet.

**Writing – original draft:** Elodie Martin, Martin Lemay, David Amarantini, Jessica Tallet.

**Writing – review & editing:** Elodie Martin, Sarah Seiwert, Lilian Fautrelle, Joseph Tisseyre, David Gasq, Martin Lemay, David Amarantini, Jessica Tallet.

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
