## [Decision Letter · Decision Letter 0]

3 Sep 2024

PONE-D-24-25640From lab to real life: Is there a link between lab-based and ecological assessment of procedural perceptual-motor learning tasks?PLOS ONE

Dear Dr. Martin,

Thank you for submitting your manuscript to PLOS ONE. After careful consideration, we feel that it has merit but does not fully meet PLOS ONE’s publication criteria as it currently stands. Therefore, we invite you to submit a revised version of the manuscript that addresses the points raised during the review process.

We look forward to receiving your revised manuscript.

Kind regards,

Rasool Abedanzadeh, Ph.D

Academic Editor

PLOS ONE

Journal Requirements:

   "This work has received financial support from the Fondation pour la Recherche en Psychomotricité et Maladies de Civilisation -  Fondation de France."

5. Please ensure that you refer to Figure 3 in your text as, if accepted, production will need this reference to link the reader to the figure.

Additional Editor Comments:

I invite the respected authors to revise and correct this manuscript according to the detailed comments of the reviewers and send it to the journal for re-evaluation.

Reviewers' comments:

Reviewer's Responses to Questions

**Comments to the Author**

1. Is the manuscript technically sound, and do the data support the conclusions?

Reviewer #1: Partly

Reviewer #2: Yes

2. Has the statistical analysis been performed appropriately and rigorously? 

Reviewer #1: Yes

Reviewer #2: Yes

3. Have the authors made all data underlying the findings in their manuscript fully available?

Reviewer #1: Yes

Reviewer #2: Yes

4. Is the manuscript presented in an intelligible fashion and written in standard English?

Reviewer #1: Yes

Reviewer #2: Yes

5. Review Comments to the Author

Reviewer #1: This study asks whether the results of experiments conducted in a controlled laboratory environment on motor skill learning can be used to predict and understand how these skills are performed in real-life everyday situations.

Answering this question is crucial for researchers and specialists working in the field of motor learning. If we can establish a link between laboratory findings and real-world performance, we can: Gain a better understanding of the motor learning process, design more effective training methods, help individuals improve their performance in various activities.

These researchers aimed to determine whether the results of simple laboratory experiments could predict an individual's performance in everyday tasks. For instance, they investigated whether a person who can effectively learn a motor sequence in a laboratory setting can also apply this skill proficiently in real-world scenarios. Their findings suggest that there is indeed a partial correlation between laboratory performance and real-world abilities. These results hold significant implications for rehabilitation professionals. This can be good, however, there are several concerns about this manuscript:

1- The introduction is long and story-like and suitable for textbooks and not scientific articles.

2- To measure motor sequence learning, it did not follow the method provided by Nissen and Bullemer and self-made assignment is used.

3- Figure 1 is completely unclear and ambiguous in terms of quality, and it is not possible to evaluate it. This problem exists in other shapes and images as well.

4- It is not clear why the general health questionnaires and Mini Mental State Examination were not used to evaluate the participants, and only self-reports were used to evaluate the participants.

5- Number of participants: The sample size of 42 participants is relatively small, especially considering the number of variables being examined. This could limit the statistical power of the study.

6- Gender imbalance: Out of 42 participants, 31 were women. This imbalance could affect the generalizability of the results.

7. Definition of "ecological": Although the "ecological" tasks are designed to reflect real-life conditions, they are still performed in a controlled laboratory environment. This might call into question their true "ecological" nature.

8. Task complexity: Some of the "ecological" tasks (such as learning sign language) may be more complex than the laboratory tasks, which could make direct comparison difficult.

9. Task order: Although the ecological tasks were presented randomly, the laboratory tasks were always performed first. This could lead to an order effect.

9. Criteria for selecting ecological tasks: More details about how the 8-person panel classified the tasks could have been useful.

These points do not necessarily indicate weaknesses in the study, but they are aspects that could be considered when interpreting the results and discussing the limitations of the study.

10- The data portion of this study has significant strengths, including comprehensiveness of data, appropriate statistical methods, and transparency in reporting. However, the relatively small sample size and the need for more detail on the handling of missing data can be considered as weaknesses.

Reviewer #2: First of all, I would like to say to the respected authors of this paper, don't get tired; This paper seems to be a valuable and good research. Nevertheless, it is recommended that the following points be considered in different sections of the papaer by respected researchers.

Abstract:

It is recommended that this section should be in accordance with the format of the journal and should be added to the type of study in the methods section and inferential statistics report in the results section.

Introduction:

Considering that the current research is considered a practical and new study, it is expected that its necessity and importance will be expressed more clearly.

Method:

It seems appropriate. It is recommended to make this section shorter and more concise if possible.

Findings: This section seems appropriate.

Discussion and conclusion:

This section needs more explanation regarding the relationship between research variables.

6. PLOS authors have the option to publish the peer review history of their article (what does this mean? ). If published, this will include your full peer review and any attached files.

**Do you want your identity to be public for this peer review?** For information about this choice, including consent withdrawal, please see our Privacy Policy .

Reviewer #1: No

Reviewer #2: No

---

## [Author Response · Author response to Decision Letter 0]

21 Oct 2024

Comments to the Author

1. Is the manuscript technically sound, and do the data support the conclusions?

Reviewer #1: Partly

Reviewer #2: Yes

2. Has the statistical analysis been performed appropriately and rigorously?

Reviewer #1: Yes

Reviewer #2: Yes

3. Have the authors made all data underlying the findings in their manuscript fully available?

Reviewer #1: Yes

Reviewer #2: Yes

4. Is the manuscript presented in an intelligible fashion and written in standard English?

Reviewer #1: Yes

Reviewer #2: Yes

5. Review Comments to the Author

Reviewer #1: This study asks whether the results of experiments conducted in a controlled laboratory environment on motor skill learning can be used to predict and understand how these skills are performed in real-life everyday situations.

Answering this question is crucial for researchers and specialists working in the field of motor learning. If we can establish a link between laboratory findings and real-world performance, we can: Gain a better understanding of the motor learning process, design more effective training methods, help individuals improve their performance in various activities.

These researchers aimed to determine whether the results of simple laboratory experiments could predict an individual's performance in everyday tasks. For instance, they investigated whether a person who can effectively learn a motor sequence in a laboratory setting can also apply this skill proficiently in real-world scenarios. Their findings suggest that there is indeed a partial correlation between laboratory performance and real-world abilities. These results hold significant implications for rehabilitation professionals. This can be good, however, there are several concerns about this manuscript:

1- The introduction is long and story-like and suitable for textbooks and not scientific articles.

We have revised the introduction to better align with the expectations of a scientific article.

2- To measure motor sequence learning, it did not follow the method provided by Nissen and Bullemer and self-made assignment is used.

The reviewer is right in the sense that our method to measure SL is adapted from the method provided by Nissen and Bullemer. For greater clarity, we added line 186: “we adapted the task initially introduced by Nissen and Bullemer (1987). Similar to their method, the stimuli appeared in four different locations. However, instead of responding by pressing a corresponding key, participants were required to use the mouse to click directly on the stimulus. The regular patterns within the sequence are acquired in a manner similar to how participants interact with a keyboard or mouse (Chambaron et al., 2008)”.

3- Figure 1 is completely unclear and ambiguous in terms of quality, and it is not possible to evaluate it. This problem exists in other shapes and images as well.

We are sorry about this point but the cause seems to be the conversion to pdf in the submission process. Indeed, the original figures which have been submitted was 2246 wide for 2446 high with a 300 dpi resolution (which perfectly answers to the submission's artwork guidelines).

We regret that the compiled submission PDF includes low-resolution preview images of the figures. We submit the revised version of our manuscript with the figures in high resolution, hoping that they will be more legible.

It is possible that the document you received includes a link at the top of the page that allows you to view the image in .tif format.

4- It is not clear why the general health questionnaires and Mini Mental State Examination were not used to evaluate the participants, and only self-reports were used to evaluate the participants.

Thank you for your comment regarding the evaluation of participants. We appreciate the importance of thorough assessment in research. However, we did not find it necessary to administer the Mini-Mental State Examination (MMSE) for our younger adult population, as the prevalence of significant cognitive impairment is generally low in this age group. Our primary goal was to balance the need for controlling variables with the feasibility of the study, particularly regarding time constraints for participants. Given these considerations, we opted to rely on self-reports and targeted inclusion criteria to ensure a sample free of neurological or psychiatric diagnoses. While we recognize that general health questionnaires and more extensive cognitive assessments could provide additional insights, we believe our approach adequately addressed the objectives of the study without overburdening participants or complicating the research process.

Furthermore, our inclusion criteria were carefully structured to recruit individuals without a history of neurological or psychiatric conditions that could impact the results. This approach aimed to ensure that our sample was as homogenous as possible concerning cognitive health. While we recognize the value of comprehensive health assessments, we believe our methodology, given the study's specific focus and constraints, was appropriate for the research objectives.

5- Number of participants: The sample size of 42 participants is relatively small, especially considering the number of variables being examined. This could limit the statistical power of the study.

We acknowledge that this relatively small sample size may limit the statistical power of our study., particularly given the number of variables being examined

We acknowledge that this relatively small sample size may limit the statistical power of our study, particularly given the number of variables being examined. We added line 604: “The limited sample size constrains the robustness of the conclusions and reduces the statistical power of the analysis, warranting careful interpretation of the results.”

6- Gender imbalance: Out of 42 participants, 31 were women. This imbalance could affect the generalizability of the results.

We appreciate your concern about the gender imbalance in our sample. We recognize that this imbalance may affect the generalizability of our findings. A quick review of the literature on the effect of gender on PPML reveals that this issue has been scarcely studied. For SA, no significant differences have been found (https://pubmed.ncbi.nlm.nih.gov/21182785/), or only among older participants (https://pubmed.ncbi.nlm.nih.gov/36617621/). For SL, we did not find any studies specifically addressing this question.

We have added a statement in the manuscript (line 607): “Even if, as far as we know, gender differences are not reported in the literature, it should be noted that our sample, is not strictly representative of the general population”.

7. Definition of "ecological": Although the "ecological" tasks are designed to reflect real-life conditions, they are still performed in a controlled laboratory environment. This might call into question their true "ecological" nature.

Our tasks are specifically designed to replicate real-life situations and contexts, allowing for the observation of skill acquisition in conditions that closely resemble those encountered in everyday life. Although the controlled environment may limit certain aspects of realism, the behaviors and tools utilized in these tasks are reflective of participants' daily experiences. Moreover, similar studies in the field have successfully employed laboratory settings to examine real-life behaviors (references 14 to 27), demonstrating that valuable insights can still be gained. We believe that the relevance of our findings to real-world applications justifies our use of the term “ecological”.

8. Task complexity: Some of the "ecological" tasks (such as learning sign language) may be more complex than the laboratory tasks, which could make direct comparison difficult.

We acknowledge that ecological tasks inherently involve more perceptual, cognitive, and motor factors than the laboratory tasks. However, despite these differences in task complexity, our hypothesis of correlation is driven by the shared learning processes.

As discussed in our manuscript (line 585), it is also possible that some correlations were not significant because of the task-specific complexity. This is the case for two tasks which do not correlate neither with SL not with SA: SL3: Learning a musical staff sequence, and SA_3: Walking without dropping a ball. As attested by the learning curves, learning process takes place for these tasks but it is possible that other cognitive or motor processes are pregnant than the procedural perceptive-motor learning process.

9. Task order: Although the ecological tasks were presented randomly, the laboratory tasks were always performed first. This could lead to an order effect.

Thank you for this comment. The laboratory tasks were intentionally performed first to establish a controlled and consistent baseline across participants. This approach allows us to minimize variability in performance due to familiarity with the tasks. In contrast, the ecological tasks involve materials and activities with which the participants are generally more familiar in their daily lives. Because they can directly perceive their success or failure in these tasks, this familiarity could have a stronger influence on their performance. By conducting the ecological tasks after the laboratory tasks, we aimed to prevent any potential carryover effects from this heightened sense of self-assessment and familiarity.

9. Criteria for selecting ecological tasks: More details about how the 8-person panel classified the tasks could have been useful.

These points do not necessarily indicate weaknesses in the study, but they are aspects that could be considered when interpreting the results and discussing the limitations of the study.

Thanks for this comment. We added clarifications in line 580: “Our ecological tasks have been classified by eight researchers and clinicians as predominantly involving either SL or SA. However, this classification may not fully account for the possibility that some tasks may integrate aspects of both types of learning. Our results suggest that this is the case given that our hypotheses are not fully validated”.

10- The data portion of this study has significant strengths, including comprehensiveness of data, appropriate statistical methods, and transparency in reporting. However, the relatively small sample size and the need for more detail on the handling of missing data can be considered as weaknesses.

Thank you for this comment. Please see our response to comment 5 and line 604-606 for the answer.

Reviewer #2: First of all, I would like to say to the respected authors of this paper, don't get tired; This paper seems to be a valuable and good research. Nevertheless, it is recommended that the following points be considered in different sections of the paper by respected researchers.

Abstract:

It is recommended that this section should be in accordance with the format of the journal and should be added to the type of study in the methods section and inferential statistics report in the results section.

Thank you for your comment. It seems to us that the journal does not appear to impose a specific format for the abstract section. However, in line with your comment, we have added line 41 “in an original research” and line 42 “with non-parametric repeated measures ANOVA, Spearman’s rank correlation tests were performed”.

Introduction:

Considering that the current research is considered a practical and new study, it is expected that its necessity and importance will be expressed more clearly.

Thank you for this comment. We have added line 137: “Although laboratory tasks have been widely used to study motor learning, their relevance to real-world skill acquisition remains debated. Ecological tasks provide a more realistic observation of how skills are acquired but their measurements require materials, time and careful quotation by trained assessors. Given the potential implications of evaluating skill learning before rehabilitation, teaching, and coaching, it is important to assess whether these two types of tasks are comparable”.

Method:

It seems appropriate. It is recommended to make this section shorter and more concise if possible.

We understand your comment but it seems difficult to condense this section without compromising clarity and understanding. We have, however, reduced the data presentation (see line 345).

Findings: This section seems appropriate.

Discussion and conclusion:

This section needs more explanation regarding the relationship between research variables.

We agree with this comment. We have added the following to line 549: “This could explain the correlations observed between learning in the SRTT task and in the sign language task, as well as between the SRTT and the knot-tying task. These tasks share the common requirement of “chunk” formation, allowing participants to perform the motor sequence more quickly and accuracy with repetition. In contrast, SA relies on mechanisms aimed at reducing sensory prediction errors (6,60). This may account for the links found between the Jump task and the Filling Cups without Spilling task, as well as between the Jump task and the Mirror Drawing task, as these tasks require sensory error correction, enabling participants to correct the error more quickly and accuracy with repetition”.

6. PLOS authors have the option to publish the peer review history of their article (what does this mean?). If published, this will include your full peer review and any attached files.

Do you want your identity to be public for this peer review? For information about this choice, including consent withdrawal, please see our Privacy Policy.

Reviewer #1: No

Reviewer #2: No

---

## [Decision Letter · Decision Letter 1]

11 Nov 2024

PONE-D-24-25640R1From lab to real life: Is there a link between lab-based and ecological assessment of procedural perceptual-motor learning tasks?PLOS ONE

Dear Dr. Martin,

Thank you for submitting your manuscript to PLOS ONE. After careful consideration, we feel that it has merit but does not fully meet PLOS ONE’s publication criteria as it currently stands. Therefore, we invite you to submit a revised version of the manuscript that addresses the points raised during the review process. Dear Authors;Please, revise your manuscript according to the reviewers' comment and then, re-submit your revised manuscript for full consideration.

We look forward to receiving your revised manuscript.

Kind regards,

Rasool Abedanzadeh, Ph.D

Academic Editor

PLOS ONE

Reviewers' comments:

Reviewer's Responses to Questions

**Comments to the Author**

1. If the authors have adequately addressed your comments raised in a previous round of review and you feel that this manuscript is now acceptable for publication, you may indicate that here to bypass the “Comments to the Author” section, enter your conflict of interest statement in the “Confidential to Editor” section, and submit your "Accept" recommendation.

Reviewer #1: (No Response)

Reviewer #2: All comments have been addressed

2. Is the manuscript technically sound, and do the data support the conclusions?

Reviewer #1: Yes

Reviewer #2: Yes

3. Has the statistical analysis been performed appropriately and rigorously? 

Reviewer #1: Yes

Reviewer #2: Yes

4. Have the authors made all data underlying the findings in their manuscript fully available?

Reviewer #1: Yes

Reviewer #2: Yes

5. Is the manuscript presented in an intelligible fashion and written in standard English?

Reviewer #1: Yes

Reviewer #2: Yes

6. Review Comments to the Author

Reviewer #1: The authors have made some revisions to the manuscript; however, the images are of low resolution and unclear. They need to be improved to ensure that readers can accurately interpret them.

Reviewer #2: HI.

The article titled " From lab to real life: Is there a link between lab-based and ecological assessment of procedural perceptual-motor learning tasks? " seems to be a valuable and practical PAPER, but it needs a major revision.

Dear authors, please check and respond to the following:

In general, the number of words in the article is very large and needs to be shortened and summarized.

1. Abstract:

The exact amount of F, P, χ², ...should be provided.

2. Keywords:

All keywords should be based on the mesh (Medical Subject Headings).

3. Introduction:

Despite the practicality of the present article, the authors have not correctly stated the importance of the necessity of the research.

The relationship between the research variables has not been properly observed.

4. Method:

Has a code of ethics been adopted for this research?

This section has been written very long.

On what basis was the number of research samples selected? Has G-power been considered?

5. Discussion and Conclusion:

The explanation of the findings in the first part of the discussion requires general editing.

It seems very necessary to present and compare the results of this research with the results of consistent and inconsistent research.

6. References:

Authors are advised to use up-to-date references.

7. PLOS authors have the option to publish the peer review history of their article (what does this mean? ). If published, this will include your full peer review and any attached files.

**Do you want your identity to be public for this peer review?** For information about this choice, including consent withdrawal, please see our Privacy Policy .

Reviewer #1: No

Reviewer #2: No

---

## [Author Response · Author response to Decision Letter 1]

24 Dec 2024

Review Comments to the Author

Reviewer #1: The authors have made some revisions to the manuscript; however, the images are of low resolution and unclear. They need to be improved to ensure that readers can accurately interpret them.

The PDF version of the manuscript currently displays lower-resolution images for convenience during the review process. However, high-resolution versions of these images are available and can be accessed by clicking on them. If the manuscript is accepted, the high-resolution images will be included in the final publication.

Reviewer #2: HI.

The article titled " From lab to real life: Is there a link between lab-based and ecological assessment of procedural perceptual-motor learning tasks? " seems to be a valuable and practical PAPER, but it needs a major revision.

Dear authors, please check and respond to the following:

In general, the number of words in the article is very large and needs to be shortened and summarized.

We have carefully revised the manuscript to improve clarity and conciseness. To preserve important details, we propose moving the description of the ecological tasks to the supplementary materials. This would reduce the word count from 8,282 to 7,605.

1. Abstract:

The exact amount of F, P, χ², ...should be provided.

Thank you for your comment. We have added this information to the abstract.

2. Keywords:

All keywords should be based on the mesh (Medical Subject Headings).

Thank you for pointing this out. There are few MeSH terms related to our topic. We included ‘Implicit Learning’ and ‘Motor Control’ as MeSH terms, but also kept the other keywords to maintain precision and capture the full scope of our study (line 56).

3. Introduction:

Despite the practicality of the present article, the authors have not correctly stated the importance of the necessity of the research.

The relationship between the research variables has not been properly observed.

Thank you for your valuable feedback. While lab-based tasks are commonly used to study motor learning, their relevance to real-world skill acquisition remains uncertain (see 7 and 42). This uncertainty presents a significant gap in the field, as it raises important questions about the ecological validity of traditional motor learning research. If lab-based findings do not accurately reflect real-world conditions, this could limit the applicability of decades of research to practical settings, such as sports and rehabilitation. In response, we have revised the introduction to clearly articulate the importance and necessity of the research. Specifically, we have emphasized the practical implications of bridging the gap between lab-based and ecological tasks and clarified the relationships between the research variables (Lines 125, 133 and, 150).

4. Method:

Has a code of ethics been adopted for this research?

Thank you for your question. The research protocol was reviewed and approved by our institutional Research Ethics Committee (CER 2020-320, Comité d’Ethique et de Recherche de l’Université Fédérale de Toulouse), ensuring that the study adhered to ethical standards. However, as this committee does not issue IRB numbers, we are unable to provide one. Nonetheless, the study was conducted in full compliance with ethical guidelines (see line 161).

This section has been written very long.

We have shortened this section while ensuring that all elements essential for the reader's understanding are retained. For the sake of transparency and reproducibility of results, we have included all methodological details but have moved some of the less critical information to the supplementary data (highlighted in green in the manuscript, lines 248 to 285).

On what basis was the number of research samples selected? Has G-power been considered?

Thank you for your comment regarding the selection of our sample size and the consideration of statistical power. The relatively small sample size has been mentioned as a limitation of the study in our discussion (line 675). However, our sample is comparable to those used in similar studies investigating perceptual-motor procedural learning, as seen in one of our previous studies (Martin et al., 2023; https://doi.org/10.1007/s10072-023-06724-w). In this review, we intentionally reported effect sizes to ensure transparency, as they provide a robust way to assess the magnitude of effects independently of the p-value, reducing reliance on statistical power for result interpretation. Post hoc analyses using G-Power indicate that with observed rho values ranging from 0.265 to 0.395, the statistical power of the study lies between 0.52 and 0.84. A common belief is that statistical power must always be at least 80% or higher. While this may be a reasonable expectation for clinical studies where significant risks or inconveniences for patients are involved, it is not always necessary in other contexts. For example, in laboratory studies or retrospective correlation analyses, a lower power may be acceptable depending on the research goals and constraints (Dorey et al., 2011; https://doi.org/10.1007/s11999-010-1435-0).

5. Discussion and Conclusion:

The explanation of the findings in the first part of the discussion requires general editing.

It seems very necessary to present and compare the results of this research with the results of consistent and inconsistent research.

Thank you for your feedback. We agree that the first part of the discussion could benefit from further editing to enhance clarity and better integrate of our findings with the existing literature. We have proposed revisions to address your concerns.

6. References:

Authors are advised to use up-to-date references.

We have reviewed our references and added the following:

Wu S, Éltető N, Dasgupta I, Schulz E. Chunking as a rational solution to the speed–accuracy trade-off in a serial reaction time task. Sci Rep. 2023 May 11;13(1):7680.

Tosatto L, Fagot J, Nemeth D, Rey A. The Evolution of Chunks in Sequence Learning. Cognitive Science. 2022;46(4):e13124.

Tosatto L, Fagot J, Nemeth D, Rey A. Chunking as a function of sequence length. Anim Cogn. 2024; 1:13.

Conway CM. How does the brain learn environmental structure? Ten core principles for understanding the neurocognitive mechanisms of statistical learning. Neuroscience & Biobehavioral Reviews. 2020 May;112:279–99.

Haar S, Sundar G, Faisal AA. Embodied virtual reality for the study of real-world motor learning. PLOS ONE. 2021 Jan 27;16(1):e0245717.

Worschech F, Passarotto E, Losch H, Oku T, Lee A, Altenmüller E. What Does It Take to Play the Piano? Cognito-Motor Functions Underlying Motor Learning in Older Adults. Brain Sci. 2024 Apr 20;14(4):405.

---

## [Decision Letter · Decision Letter 2]

13 Jan 2025

PONE-D-24-25640R2From lab to real life: Is there a link between lab-based and ecological assessment of procedural perceptual-motor learning tasks?PLOS ONE

Dear Dr. Martin,

Thank you for submitting your manuscript to PLOS ONE. After careful consideration, we feel that it has merit but does not fully meet PLOS ONE’s publication criteria as it currently stands. Therefore, we invite you to submit a revised version of the manuscript that addresses the points raised during the review process. Dear Authors,According to first reviewer's comment the manuscript need major revise and I am agree with him.Please revise your manuscript carefully and then re-submit for full consideration.

We look forward to receiving your revised manuscript.

Kind regards,

Rasool Abedanzadeh, Ph.D

Academic Editor

PLOS ONE

Reviewers' comments:

Reviewer's Responses to Questions

**Comments to the Author**

1. If the authors have adequately addressed your comments raised in a previous round of review and you feel that this manuscript is now acceptable for publication, you may indicate that here to bypass the “Comments to the Author” section, enter your conflict of interest statement in the “Confidential to Editor” section, and submit your "Accept" recommendation.

Reviewer #1: All comments have been addressed

Reviewer #2: (No Response)

2. Is the manuscript technically sound, and do the data support the conclusions?

Reviewer #1: Yes

Reviewer #2: (No Response)

3. Has the statistical analysis been performed appropriately and rigorously? 

Reviewer #1: Yes

Reviewer #2: (No Response)

4. Have the authors made all data underlying the findings in their manuscript fully available?

Reviewer #1: Yes

Reviewer #2: (No Response)

5. Is the manuscript presented in an intelligible fashion and written in standard English?

Reviewer #1: Yes

Reviewer #2: (No Response)

6. Review Comments to the Author

Reviewer #1: The images of charts are completely unclear, and the authors have not made the necessary revisions. A further review is required.

Reviewer #2: (No Response)

7. PLOS authors have the option to publish the peer review history of their article (what does this mean? ). If published, this will include your full peer review and any attached files.

**Do you want your identity to be public for this peer review?** For information about this choice, including consent withdrawal, please see our Privacy Policy .

Reviewer #1: No

Reviewer #2: No

---

## [Author Response · Author response to Decision Letter 2]

20 Jan 2025

Review Comments to the Author

Reviewer #1: The images of charts are completely unclear, and the authors have not made the necessary revisions. A further review is required.

Thank you for your valuable feedback. In response to your comments, we have increased the resolution of the chart images in the manuscript. We hope that this improvement makes the figures clearer and more readable.

---

## [Decision Letter · Decision Letter 3]

6 Feb 2025

From lab to real life: Is there a link between lab-based and ecological assessment of procedural perceptual-motor learning tasks?

PONE-D-24-25640R3

Dear Dr. Martin,

We’re pleased to inform you that your manuscript has been judged scientifically suitable for publication and will be formally accepted for publication once it meets all outstanding technical requirements.

Kind regards,

Rasool Abedanzadeh, Ph.D

Academic Editor

PLOS ONE

Additional Editor Comments (optional):

Reviewers' comments:

Reviewer's Responses to Questions

**Comments to the Author**

1. If the authors have adequately addressed your comments raised in a previous round of review and you feel that this manuscript is now acceptable for publication, you may indicate that here to bypass the “Comments to the Author” section, enter your conflict of interest statement in the “Confidential to Editor” section, and submit your "Accept" recommendation.

Reviewer #1: All comments have been addressed

2. Is the manuscript technically sound, and do the data support the conclusions?

Reviewer #1: Yes

3. Has the statistical analysis been performed appropriately and rigorously? 

Reviewer #1: Yes

4. Have the authors made all data underlying the findings in their manuscript fully available?

Reviewer #1: Yes

5. Is the manuscript presented in an intelligible fashion and written in standard English?

Reviewer #1: Yes

6. Review Comments to the Author

Reviewer #1: (No Response)

7. PLOS authors have the option to publish the peer review history of their article (what does this mean? ). If published, this will include your full peer review and any attached files.

**Do you want your identity to be public for this peer review?** For information about this choice, including consent withdrawal, please see our Privacy Policy .

Reviewer #1: No

---

## [Editor Report · Acceptance letter]

PONE-D-24-25640R3

PLOS ONE

Dear Dr. Martin,

I'm pleased to inform you that your manuscript has been deemed suitable for publication in PLOS ONE. Congratulations! Your manuscript is now being handed over to our production team.

Kind regards,

on behalf of

Dr. Rasool Abedanzadeh

Academic Editor

PLOS ONE